# Eriodictyol Suppresses Gastric Cancer Cells via Inhibition of PI3K/AKT Pathway

**DOI:** 10.3390/ph15121477

**Published:** 2022-11-27

**Authors:** Hui Shan, Xin Zhang, Yalu Mi, Jihui Jia, Bo Wang, Qing Yang

**Affiliations:** 1Institute of Pathogen Biology, School of Basic Medical Sciences, Cheeloo College of Medicine, Shandong University, Jinan 250012, China; 2Department of General Surgery, Qilu Hospital, Cheeloo College of Medicine, Shandong University, Jinan 250012, China; 3College of Traditional Chinese Medicine, Shandong University of Traditional Chinese Medicine, Jinan 250355, China; 4Shandong Key Laboratory of Infection and Immunity, Shandong University, Jinan 250012, China; 5Shandong University-Karolinska Institute Collaborative Laboratory for Cancer Research, Shandong University, Jinan 250012, China; 6Department of Traditional Medicine, Qilu Hospital, Cheeloo College of Medicine, Shandong University, Jinan 250012, China

**Keywords:** eriodictyol, *Polygoni orientalis* Fructus, gastric cancer, apoptosis, PI3K/AKT, network pharmacology, molecular docking

## Abstract

Gastric cancer (GC) is among the five most common malignancies worldwide. Traditional chemotherapy cannot efficiently treat the disease and faces the problems of side effects and chemoresistance. *Polygoni orientalis* Fructus (POF), with flavonoids as the main bioactive compounds, exerts anti-cancer potential. In this study, we compared the anti-GC effects of the main flavonoids from POF and investigated the anti-cancer effects of eriodictyol towards GC both in vitro and in vivo. CCK-8 assays were performed to examine the inhibitory effects of common flavonoids from POF on GC cell viability. Colony formation assays were used to determine cell proliferation after eriodictyol treatment. Cell cycle distribution was analyzed using flow cytometry. Induction of apoptosis was assessed with Annexin V/PI staining and measurement of related proteins. Anti-cancer effects in vivo were investigated using a xenograft mouse model. Potential targets of eriodictyol were clarified by network pharmacological analysis, evaluated by molecular docking, and validated with Western blotting. We found that eriodictyol exhibited the most effective inhibitory effect on cell viability of GC cells among the common flavonoids from POF including quercetin, taxifolin, and kaempferol. Eriodictyol suppressed colony formation of GC cells and induced cell apoptosis. The inhibitory effects of eriodictyol on tumor growth were also validated using a xenograft mouse model. Moreover, no obvious toxicity was identified with eriodictyol treatment. Network pharmacology analysis revealed that PI3K/AKT signaling ranked first among the anti-GC targets. The molecular docking model of eriodictyol and PI3K was constructed, and the binding energy was evaluated. Furthermore, efficient inhibition of phosphorylation and activation of PI3K/AKT by eriodictyol was validated in GC cells. Taken together, our results identify eriodictyol as the most effective anti-GC flavonoids from POF and the potential targets of eriodictyol in GC. These findings suggest that eriodictyol has the potential to be a natural source of anti-GC agents.

## 1. Introduction

Gastric cancer (GC) ranks as the sixth most common cancer in incidence and the third in mortality among all the malignancies worldwide [1]. As most patients are diagnosed at advanced stages, surgical resection alone of primary tumor and regional lymph nodes usually cannot prevent progress of GC successfully [2]. To overcome the limitations of surgery, chemotherapy is usually applied perioperatively and after surgery. In addition, multiple targeted chemotherapies have been developed and are often used in combination with traditional chemotherapy [3,4,5]. Platinum-based chemotherapy is still used as first-line chemotherapy for patients, especially those with advanced GC. Although it is reported that almost 60% of GC patients respond to available chemotherapy, most patients experience severe side effects and develop chemoresistance eventually [5,6]. Therefore, it is urgent to develop new efficient drugs to treat GC patients.

An increasing amount of evidence has shown that traditional Chinese medicine is a promising source for identifying new agents in cancer prevention and treatment. *Polygoni orientalis* Fructus (POF), the dried ripe fruit of *Polygonum orientale* L., is a well-known medicinal plant and possesses hepatoprotective and anti-cancer activities [7,8]. In POF, the main bioactive compounds are flavonoids, which include quercetin, taxifolin, kaempferol, and eriodictyol. Among these natural flavonoids, the anti-cancer capabilities of quercetin, taxifolin, and kaempferol have been extensively investigated. The chemo-preventive effect of eriodictyol was also identified. In a chemical carcinogen 1,2-dimethylhydrazine-induced animal model of colon cancer, through the antioxidant defense mechanism, eriodictyol decreased lipid peroxidation levels and inhibited preneoplastic lesions [9]. This indicates that eriodictyol can be applied in cancer treatment.

In the present study, we compared the inhibitory effects of eriodictyol on cell viability with the three other flavonoids from POF, namely quercetin, taxifolin, and kaempferol. The data demonstrated that eriodictyol inhibited cell viability most effectively among these common flavonoids. Then, anti-GC effects of eriodictyol were further examined both in vitro and in vivo. Safety of eriodictyol treatment in the xenograft mouse model was also evaluated. Finally, the potential targets of eriodictyol to inhibit GC cells were revealed using network pharmacological analysis, evaluated by molecular docking, and validated experimentally. Our results identified the inhibitory role of eriodictyol in GC and the underlying mechanism, supporting eriodictyol as a potential natural compound for chemotherapy in GC.

## 2. Results

### 2.1. Eriodictyol Suppresses Cell Viability of GC Cells

The main flavonoids isolated from POF include quercetin, taxifolin, kaempferol, and eriodictyol. The first three of them have been identified as being able to suppress GC efficiently [10,11,12]. Therefore, in this study, we first compared cell inhibition capability of eriodictyol in GC cells with that of quercetin, taxifolin, and kaempferol. CCK-8 assays were performed and the results showed that eriodictyol exhibited the most efficient inhibitory effects among these examined common flavonoids from POF in AGS and HGC-27 cells at different concentrations and different time points (Figure 1A). Then, inhibition of cell viability by eriodictyol in MKN-45 cells was confirmed (Figure 1B). In addition, the cytotoxic effect of eriodictyol on human gastric epithelial GES-1 cells was much weaker than that on GC cells (Figure 1C). This indicated that eriodictyol displayed cytotoxicity selectively against GC cells. Moreover, IC 50 of eriodictyol for the above cells at 24, 48, and 72 h was calculated according to the results of CCK-8 assays (Table 1). In the following experiments, the concentration (150 µM) approximately equivalent to IC 50 at 48 h and two lower concentrations (100 and 50 µM) were applied to AGS and HGC-27 cells. Because IC 50 of eriodictyol for MKN-45 cells is higher than that of the other two GC cells, different concentrations (100, 200, and 300 µM) of eriodictyol were used. Our results demonstrated that eriodictyol could inhibit cell viability efficiently, and the inhibition of GC cells induced by eriodictyol was dose- and time-dependent.

### 2.2. Eriodictyol Inhibits Cell Proliferation of GC Cells

To further examine the inhibitory effects of eriodictyol on proliferation of GC cells, colony formation assays were performed. Eriodictyol treatment significantly reduced the foci numbers as well as sizes in GC cells (Figure 2A,B). The inhibitory effects of eriodictyol on colony formation are dose-dependent in line with those on cell viability. Then, cell cycle analysis was performed, and an obvious increase in cell fractions in the sub-G1 phase was observed, indicating induction of apoptosis (Figure 2C,D). Moreover, there is a gradual decrease in cell population in the G2/M phase with the increase in eriodictyol concentration (Figure 2C,D). Our results suggest the anti-proliferation effects of eriodictyol on GC cells.

### 2.3. Eriodictyol Induces Cell Apoptosis in GC Cells

Further experiments were performed to confirm the effects of eriodictyol on induction of cell apoptosis. Results of flow cytometry indicated that percentages of Annexin V-positive cells increased gradually when cells were treated with increasing concentrations of eriodictyol (Figure 3A,B). This suggests that proportions of cells in both the early and the late apoptotic stages increase when treated with eriodictyol. Then, induction of apoptosis in GC cells was investigated with Western blot examination of apoptotic executive proteins. A substantial accumulation of cleaved PARP-1 and Caspase-3 was observed in GC cells due to eriodictyol treatment (Figure 3C,D). These data demonstrate that eriodictyol induces cell apoptosis in GC cells.

### 2.4. Eriodictyol Inhibits Tumor Growth In Vivo

To determine the anti-cancer efficacy of eriodictyol in vivo, nude mice were used, and MKN-45 cells were inoculated subcutaneously to establish xenograft tumor models. Cisplatin, as the first-line chemotherapeutic agent, was used as the positive control. All of the nude mice survived to the end of xenograft experiments. Eriodictyol treatment significantly suppressed tumor growth as illustrated by smaller tumor volumes and weights (Figure 4A,B,D). Eriodictyol efficiently inhibited xenografts growth at both concentrations. In our tests, eriodictyol at higher concentration exhibited similar inhibitory efficiency to cisplatin. In addition, eriodictyol treatment significantly decreased Ki-67-positive cells as visualized by IHC staining, indicating that eriodictyol inhibits proliferation of GC cells in vivo (Figure 4D,E). Then, safety of eriodictyol treatment was evaluated. No obvious differences in body weights of nude mice were observed when treated with eriodictyol at both concentrations, while a substantial decrease in body weights was identified in cisplatin-treated mice (Figure 4C). H&E staining indicated that eriodictyol did not cause visible damage to major organs including hearts, livers, and kidneys (Figure 4F). Moreover, eriodictyol treatment did not make detectable changes in serum levels of AST and ALT, two major enzymes reflexing the liver function (Figure 4G). Although no obvious changes were found in gross specimens from cisplatin-treated mice, hepatocyte edema was revealed in two of six mice (Figure 4F). An increase in serum levels of AST and ALT was also identified in mice treated with cisplatin (Figure 4G). Taken together, these results demonstrated that eriodictyol exerts potent anti-cancer capability in GC and is safe at the treatment dosage in vivo.

### 2.5. Network Pharmacology Prediction of the Possible Pathways Regulated by Eriodictyol

To explore the mechanism underlying inhibitory effects of eriodictyol on GC cells, we used network pharmacology to analyze the potential targets of eriodictyol in GC. A total of 200 drug-associated targets and 11,261 disease-related targets were identified with 131 overlapping potential targets obtained (Figure 5A). Then, the overlapping targets were imported into the STRING database, and with the minimum required interaction score set as 0.7, 100 enriched targets were obtained and used to construct a PPI network. In this network, the first three core targets were HSP90AA1, AKT1, and SRC (Figure 5B). Both PIK3CG and AKT1, the key components in the PI3K/AKT pathway, are included in the main nodes of the network (Figure 5C).

Then, the overlapping targets from the drug of eriodictyol and the disease of GC were used to perform GO and KEGG analysis. The first 10 significantly enriched terms of the biological process, cellular component, and molecular function are shown in Figure 5D. The kinase activity is the main molecular function item that is affected by eriodictyol treatment. Furthermore, the results of KEGG analysis revealed that the PI3K/AKT signaling pathway ranked first among the 20 top terms (Figure 5E). These data indicate that eriodictyol inhibits GC cell mainly by intervening in the PI3K/AKT pathway.

### 2.6. Molecular Docking Assessment

Then, molecular docking was performed to confirm the binding modes of eriodictyol with PIK3CG (PDB ID: 1E8X). The results revealed that eriodictyol was docked in a similar position to ATP. As shown in Figure 6A,B, eriodictyol formed five hydrogen bonds with CYS-275, GLU-302, HIS-304, and GLU-826 in PIK3CG. In addition, the binding energy was calculated to evaluate the degree of complementarity between the component (eriodictyol or ATP) and the PIK3CG protein. The binding energies between eriodictyol and PIK3CG and between ATP and PIK3CG were −5.98 and −3.17 kJ/mol, respectively (Figure 6C). As the binding energy between eriodictyol and PIK3CG was lower than that between ATP and PIK3CG, our results indicate a more stable conformation between eriodictyol and PIK3CG. These suggest that eriodictyol could compete with ATP, bind to the active sites of PIK3CG, and thus inhibit the activity of PI3K kinase.

### 2.7. Eriodictyol Inhibits GC Cells through PI3K/AKT Pathway

PI3K plays a critical role in transmitting oncogenic signals by regulating AKT in GC. Therefore, with bioinformatics prediction of PI3K as the main target of eriodictyol in GC, effects of eriodictyol on the PI3K/AKT pathway were evaluated with Western blotting. As shown in Figure 7, in GC cells, the phosphorylation (p85 Tyr458/p55 Tyr199) of PI3K was obviously decreased by eriodictyol treatment in a dose-dependent manner. Meanwhile, treatment with eriodictyol also evidently reduced phosphorylation (Ser473) of AKT. Our results suggest that eriodictyol treatment inhibited phosphorylation of PI3K and AKT, leading to inactivation of the PI3K/AKT pathway in GC.

## 3. Discussion

GC is the third leading cause of cancer-related deaths worldwide [1]. Late diagnosis and limited chemotherapy options account for the high mortality of GC [3,13]. Although the efficacy of chemotherapy in GC patients is improving, chemoresistance and severe side effects are noted in clinical practice [13,14]. As the 5-year survival rate is less than 10% in advanced GC patients, it is critical to find new anti-cancer drugs for GC patients. In recent years, an increasing amount of evidence suggests that traditional Chinese medicine exhibits efficient anti-cancer capability and is widely used to treat malignancies including GC [15,16]. POF is a traditional herb and has a long history in treating various diseases [17,18]. Previous research work has identified flavonoids as the main and bioactive components of POF. Studies applying combined liquid chromatography and mass spectrometry technology revealed that flavonoids from POF consist of quercetin, taxifolin, kaempferol, and eriodictyol [8,17,18,19].

Quercetin, taxifolin, and kaempferol are all well-studied flavonoids with established anti-cancer abilities in various malignancies including GC [10,11,12]. To compare the inhibitory effects of eriodictyol with the above three flavonoid compounds, we performed CCK-8 assays. The results showed that among these flavonoids, eriodictyol exhibited the best suppressive ability on cell viability in the different GC cancer cells we examined. However, this is not the case in other malignancies. When eriodictyol was applied in melanoma, contrary results from no, moderate, to strong inhibitory effects were presented by different studies [20,21,22]. The inhibitory effects of eriodictyol are comparable to those of quercetin [23]. In lung cancer and leukemia, IC50 of eriodictyol is comparable to that of quercetin and much lower than that of kaempferol [24]. These studies including our data indicate that flavonoid compounds differ in their suppressive capacities in different cancers. Eriodictyol inhibits GC most efficiently among the flavonoids from POF we explored in this study.

Eriodictyol is a common and widespread flavonoid compound with established antioxidant capacities and protective role in neurological and cardiological diseases [25,26,27,28]. As antioxidants have attracted attention as potential anti-cancer agents, the anti-cancer effects of eriodictyol were evaluated [28]. Eriodictyol was found to decrease chemical carcinogen-induced precancerous lesion in colon and suppress the malignant progression of colorectal cancer cells [9,29]. In mice epithelial cells JB6Cl41, eriodictyol treatment suppresses EGF-induced anchorage-independent colony formation in soft agar, which indicates the inhibitory role of eriodictyol in malignant transformation [30]. Then, the anti-cancer capabilities of eriodictyol were revealed in glioma, lung cancer, and nasopharyngeal cancer [31,32,33]. Nevertheless, the role of eriodictyol in GC has not been explored. In this study, we demonstrated that eriodictyol significantly inhibited cell viability of GC cells, and the inhibitory effect was time- and dose-dependent. Our results indicate that eriodictyol may exert anti-cancer effects on GC. Moreover, we performed further experiments and validated that eriodictyol can inhibit growth of GC cells both in vitro and in vivo.

Besides chemoresistance, another challenge for clinical application of traditional chemotherapy is side effects, especially toxicity to the kidney, liver, and heart [34,35]. Therefore, when seeking new therapeutic agents, it is important to assess the safety of the new drug. The cytotoxic effect of eriodictyol is selective to GC cells and not gastric epithelial cells, which is a promising finding. Moreover, eriodictyol exhibited obvious anti-cancer efficacy in a xenograft mouse model without significant impact on the body weight or obvious toxicity to the critical organs of the treated mice. Consistent with our results, when treating glioma in xenografted mice, eriodictyol was shown to inhibit tumor growth but did not reduce the body weights of model mice [31]. However, the major organs were not examined histologically in that study. To our knowledge, this is the first evaluation of the safety of eriodictyol application in treating malignancy with histological examination and using blood biochemical indexes. For the dosage of eriodictyol, He et al. used 100mg/kg eriodictyol in the mouse model for 45 days, with the total amount being more than that used in our study [25]. The usage of eriodictyol exerted protective effects in lipopolysaccharide-triggered neuroinflammation without obvious influence of food intake and body weights of the treated mice. Furthermore, although MKN-45 [36], the cell line with a higher IC50, was chosen for the xenograft model, the dosage we used in mice is efficient and most importantly, safe. Therefore, our results suggest the promising application of eriodictyol in clinical practice.

The concept of “network pharmacology” was first proposed in 2007 and developed rapidly with advances in bioinformatics and system biology [37,38]. Network pharmacology has become a branch of pharmacology which is often used to reveal the relationship among drugs, diseases, and targets. Network pharmacology is quite different from the conventional “one disease–one target–one drug” pharmacological research strategies and is widely used in identification of natural bioactive compounds. In this study, we determined the targets of eriodictyol and GC and using the overlapping targets obtained a core PPI network. Both PIK3CG and AKT1 are included in the network, and the PI3K/AKT signaling pathway ranks the first in the following KEGG analysis. According to the results of network pharmacology, PI3K is identified as the promising target of eriodictyol related to GC.

Flavonoids are known to compete for the ATP-binding sites of protein kinases and phospholipid kinases and inhibit kinase activities [39,40,41]. Through competing with ATP, quercetin and several other flavonoids work as efficient inhibitors of inositol polyphosphate kinase [40]. Kaempferol was shown to inhibit p21 activating kinase 4 in breast cancer [39]. With a similar structure to kaempferol, eriodictyol was revealed as the inhibitor of ribosomal S6 kinase 2 and suppressed neoplastic transformation in mice epithelial cells [30]. In macrophages, through interacting with the ATP-binding sites of Jun-N terminal kinase, eriodictyol inhibited kinase activity and exerted anti-inflammation potential [42]. After predicting PI3K as the most promising target of eriodictyol, we performed molecular docking to propose a binding model. The docking result indicated that eriodictyol interacted with the ATP-binding site of PI3KCG and possessed lower binding energy and more stable conformation when compared with ATP.

Aberrant activation of the PI3K/AKT pathway has been reported in a variety of malignancies including GC [43]. Once activated by receptor tyrosine kinases, PI3K binds to plasma membrane and transforms phosphatidylinositol-4,5-bisphosphate (PIP2) into phosphatidylinositol-3,4,5-trisphosphate (PIP3). As the secondary messenger, PIP3 interacts with and phosphorylates AKT1, resulting in activation of growth and cell survival. Due to the crucial role of PI3K/AKT signaling in cancer, several inhibitors targeting this pathway have been developed or are under development [44,45]. In this study, after network pharmacology and molecular docking analysis, we performed in vitro validation experiments and confirmed that eriodictyol can efficiently inhibit phosphorylation and activation of PI3K/AKT signaling. Our study not only reveals the anti-cancer potential of eriodictyol but also identifies the main targets of eriodictyol in GC.

## 4. Materials and Methods

### 4.1. Cell Culture and Reagents

Three human GC cell lines, namely AGS, HGC-27, and MKN-45, and human gastric mucosal epithelial GES-1 cells were used in this study. Cells were obtained from the Cell Resource Center, Institute of Biochemistry and Cell Biology at the Chinese Academy of Science (Shanghai, China). AGS cells were cultured in F12 medium (Macgene, Beijing, China) plus 12% (*v*/*v*) fetal bovine serum (FBS) (Biological Industries, Cromwell, CT, USA). HGC-27, MKN-45, and GES-1 cells were cultured in RPMI-1640 medium (Biological Industries, USA) plus 10% (*v*/*v*) FBS. The cell lines were all incubated at 37 °C in a humidified atmosphere with 5% CO_2_. Eriodictyol (Cat#T6S0232, purity: 98.1%), quercetin (Cat#T2174, purity: 98%), taxifolin (Cat#T1738, purity: 94.91%), and kaempferol (Cat#T2177, purity: 99.41%) were purchased from TargetMol (Wellesley Hills, MA, USA).

### 4.2. CCK-8 Assays

Cells were seeded in 96-well plates at a density of 5000 cells/well the day before assays. A series of concentrations (0, 50, 100, 150, 200, 250 μM for AGS and HGC-27 cells; 0, 100, 200, 300, 400, 500 μM for MKN-45 and GES-1 cells) of eriodictyol were added to the cells and incubated for 24, 48, and 72 h. Afterwards, the treated cells were incubated with 100 μL CCK-8 reagent solution (MedChemExpress LLC, Monmouth Junction, NJ, USA) for an additional 2 h at 37 °C. The absorbance was measured at 450 nm with 650 nm as a reference using an Infinite M200 spectrophotometer (Tecan, Grödig, Austria). Assays were performed in triplicate. The IC50 value was calculated using GraphPad Prism 8.0.2 software (GraphPad Software, La Jolla, CA, USA).

### 4.3. Colony Formation Assays

GC cells were plated in six-well plates at a density of 500 cells/well. After 16 h, cells were treated with a series of concentrations (0, 50, 100, 150 μM for AGS and HGC-27 cells; 0, 100, 200, 300 μM for MKN-45 cells) of eriodictyol and then incubated at 37 °C for 10–14 days. Then, the cell colonies were washed with PBS, fixed with methanol, and subsequently stained with crystal violet. Finally, the cell colonies were photographed, and those with more than 50 cells were counted. The experiments were performed in triplicate.

### 4.4. Cell Apoptosis Assays

Cell apoptosis was measured according to the manual of FITC Annexin V Apoptosis Detection Kit I (BD Biosciences, San Jose, CA, USA). Briefly, the GC cells were plated in six-well plates at a density of 4×10^5^ cells/well and cultured overnight. After treatment with different concentrations of eriodictyol for 48 h, the cells were collected, washed with cold PBS, and incubated with 5 μL of FITC Annexin V and 5 μL of propidium iodide (PI) for 15min at room temperature in the dark. Then, the samples were examined using a CytoFLEX flow cytometer (Beckman Coulter, Atlanta, GA, USA). The experiments were repeated three times.

### 4.5. Cell Cycle Analysis

GC cells were plated in six-well plates at a density of 4 × 10^5^ cells/well. After treatment with eriodictyol for 48 h, cells were collected, washed with cold PBS, and fixed with 70% ethanol at 4 °C overnight. Then, the cells were treated with PI (Beyotime, Shanghai, China) and analyzed using a CytoFLEX flow cytometer. Cell cycle analysis was performed using Flow Jo 10 software (Becton Dickinson, NJ, USA). The experiments were repeated three times.

### 4.6. Western Blot

After treatment with eriodictyol for 48 h, cells were collected to extract total protein with RIPA lysis buffer (Beyotime, China). The protein concentration was measured with the BCA Protein Assay Kit (Beyotime, China). The proteins were separated via SDS-PAGE and transferred to nitrocellulose membrane (Merck Millipore, Darmstadt, Germany). The membranes were incubated with specific primary antibodies against PI3K (1:1000; Cell Signaling Technology, #4257), p-PI3K (1:1000; Cell Signaling Technology, #4228), AKT (1:1000; Cell Signaling Technology, #9272), p-AKT (1:1000; Cell Signaling Technology, #4060), PARP-1 (1:1000; Cell Signaling Technology, #9542), Caspase-3 (1:1000; Cell Signaling Technology, #9662), and β-actin (1:1000; Cell Signaling Technology,#4967). Then, the protein bands were visualized as previously described [46].

### 4.7. Xenograft Tumor Model

Five-week-old male BALB/c nude mice were purchased from Beijing Vital River Laboratory Animal Technology (Beijing, China). MKN-45 cells (4 × 10^5^) were resuspended in 200 μL PBS and injected subcutaneously into the flank region of each mouse. When the tumor volumes (V = W^2^ × L/2) reached 100 mm^3^ (regarded as day 0), the mice were randomly assigned to four groups (*n* = 6 for each group). In Group I, mice were injected intraperitoneally with 0.9% NaCl every 2 days; in group II, mice were injected intraperitoneally with cisplatin (5 mg/kg) every 5 days; in group III, mice were injected intraperitoneally with eriodictyol (100mg/kg) every 2 days; and in group IV, mice were injected intraperitoneally with eriodictyol (200 mg/kg) every 2 days [31]. Tumor volumes were measured every 2 days, and body weights of mice were measured every 5 days. After three weeks, the mice were euthanized. The tumors and major organs including hearts, livers, and kidneys were collected. The blood serum of mice was also collected to determine levels of AST and ALT using a BS-240VET Auto Chemistry Analyzer (Mindray, Shenzhen, China).

### 4.8. Hematoxylin and Eosin (H&E) and Immunohistochemistry (IHC) Staining

The collected organs and tumor tissues were fixed in 4% paraformaldehyde overnight. Then, after dehydration and embedding in paraffin, the tissue sections of hearts, livers, and kidneys were stained with H&E to evaluate the major organs histologically. The tumor sections were used for IHC staining. For IHC staining, the sections were incubated with anti-K-i67 antibody (1:200, Abcam, ab92742). The DAB staining and analysis of K-i67-positive cells were performed as previously described [46].

### 4.9. Network Pharmacological Analysis

The targets of eriodictyol were collected from the following three databases: (1) Traditional Chinese Medicine Systems Pharmacology (TCMSP, https://old.tcmsp-e.com/tcmsp.php, accessed on 25 February 2022), (2) Swiss Target Prediction (http://www.swisstargetprediction.ch/, accessed on 13 February 2022), (3) PharmMapper (http://www.lilab-ecust.cn/pharmmapper, accessed on 25 February 2022). The disease targets named “gastric cancer” were obtained from the following 5 databases: (1) GeneCards (https://www.genecards.org, accessed on 9 February 2022), (2) Online Mendelian Inheritance in Man (OMIM, https://www.omim.org/, accessed on 9 February 2022), (3) DrugBank (https://www.drugbank.ca/, accessed on 9 February 2022), (4) Therapeutic Target Database (TTD, http://db.idrblab.net/ttd/, accessed on 9 February 2022), (5) PharmGKB (https://www.pharmgkb.org/, accessed on 9 February 2022). Then, drug targets and disease targets were crossed to obtain potential targets of eriodictyol for the treatment of GC. The gene symbols of potential targets were uploaded to STRING (https://string-db.org/, accessed on 21 April 2022) to construct a protein–protein interaction (PPI) network. The organism was set to “Homo sapiens”, and the minimum required interaction score was set as 0.7. The resultant PPI network was imported into and visualized in Cytoscape 3.9.1 software (Cytoscape Consortium, Seattle, WA, USA). In addition, further Gene Ontology (GO) and Kyoto Encyclopedia of Genes and Genomes (KEGG) analyses were performed for the key targets obtained from the PPI network. The *p*-value cutoff was set as 0.01. The first 10 processes and pathways of GO, including molecular function, biological process, and cellular component, and the first 20 KEGG pathways were selected to analyze the potential target pathways of eriodictyol for the treatment of GC.

### 4.10. Molecular Docking

The structure of eriodictyol was derived from PubChem website and optimized with Chem3D 2014 software. The PDB file of 3D crystal structures for PIK3CG (1E8X, ligand: adenosine triphosphate, ATP) was downloaded from the Protein Data Bank (https://www.rcsb.org/, accessed on 17 September 2022). Molecular docking was performed using Autodock software, the binding energy was evaluated, and the docking result was visualized with PyMOL and LigPlot+ tools.

### 4.11. Statistical Analysis

The experimental data are presented as means ± SD. The data were analyzed with GraphPad Prism 8.0.2 software (GraphPad Software, La Jolla, CA, USA) using Student’s *t*-test or one-way ANOVA test. *p* < 0.05 was considered to indicate statistical significance.

## 5. Conclusions

In summary, the present study demonstrated that eriodictyol exhibits the most effective inhibitory effect on cell viability of GC cells among the common flavonoids in POF, which include quercetin, taxifolin, kaempferol, and eriodictyol. Eriodictyol suppresses proliferation of GC cells, induces cell apoptosis, and inhibits tumor growth in vivo. PI3K/AKT signaling ranks first among the anti-GC targets of eriodictyol and was inhibited efficiently by eriodictyol treatment in GC cells (Figure 8). This provides insights into clinical application of eriodictyol in GC treatment.

## Figures and Tables

**Figure 1 pharmaceuticals-15-01477-f001:**
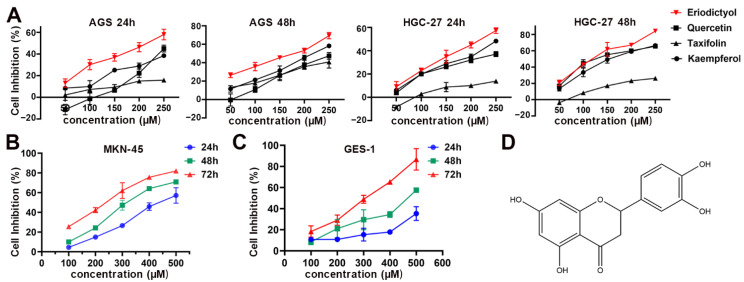
Inhibition of cell viability of gastric cancer (GC) cells by eriodictyol, quercetin, taxifolin, and kaempferol. (**A**) Cell inhibition by eriodictyol, quercetin, taxifolin, and kaempferol of AGS and HGC-27 cells was determined by CCK-8 assays. (**B**) Cell inhibition by eriodictyol of MKN-45 cells was determined by CCK-8 assays. (**C**) Cell inhibition by eriodictyol of GES-1 cells was determined by CCK-8 assays. (**D**) The chemical structure of eriodictyol. Three independent experiments were performed, and data are presented as mean ± SD.

**Figure 2 pharmaceuticals-15-01477-f002:**
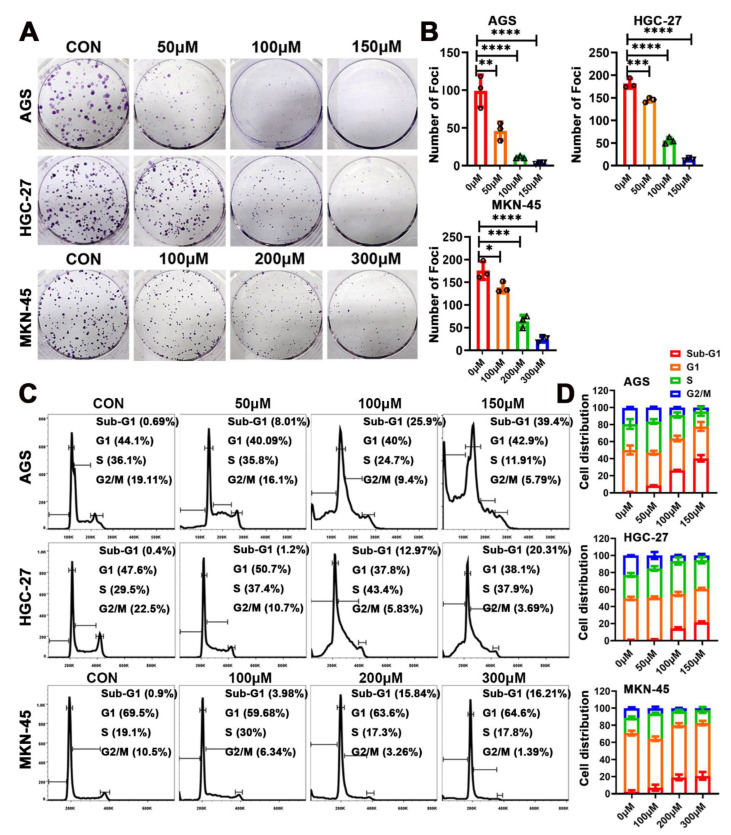
Eriodictyol inhibits cell proliferation of GC cells. (**A**) The representative graphs of colony formation assays are shown. (**B**) The quantitative analysis of colony formation assays is demonstrated as a histogram. (**C**) The cell cycle distribution was analyzed using flow cytometry, and the representative results are shown. (**D**) The quantitative analysis of cell cycle distribution is demonstrated as a histogram. Three independent experiments were performed, and data are presented as mean ± SD. * *p* < 0.05, ** *p* < 0.01, *** *p* < 0.001, **** *p* < 0.0001.

**Figure 3 pharmaceuticals-15-01477-f003:**
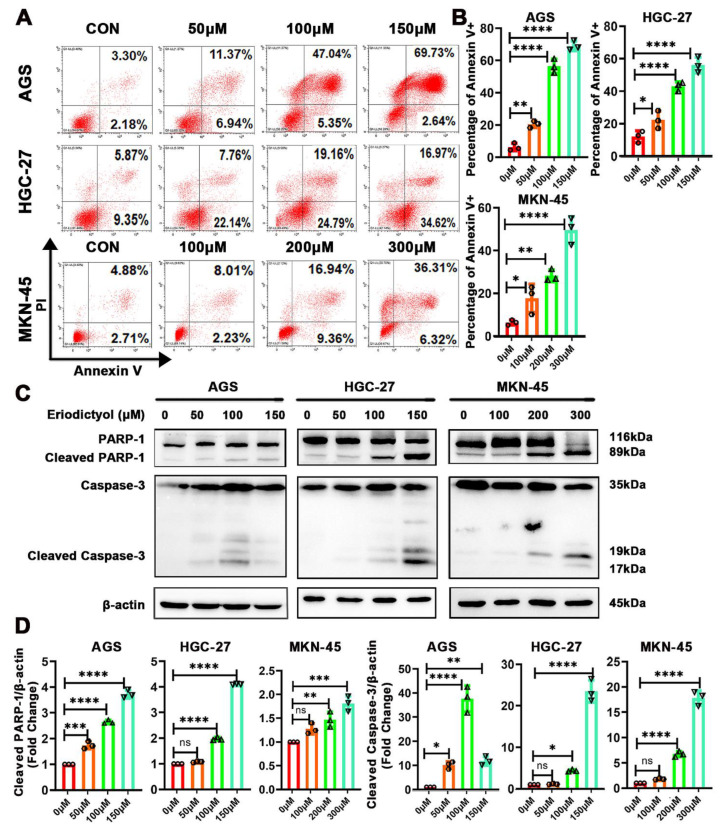
Eriodictyol induces cell apoptosis in GC cells. (**A**) Cells in early and late apoptotic stages were detected with Annexin V/PI stain and flow cytometry analysis. (**B**) The quantitative analysis of Annexin V-positive percentages is demonstrated as a histogram. (**C**) The levels of PARP-1, cleaved PARP-1, Caspase-3, and cleaved Caspase-3 were measured with Western blotting. (**D**) The relative protein levels of cleaved PARP-1 and cleaved Caspase-3 were quantified and are demonstrated as a histogram. Three independent experiments were performed, and data are presented as mean ± SD. ns: no significance, * *p* < 0.05, ** *p* < 0.01, *** *p* < 0.001, **** *p* < 0.0001.

**Figure 4 pharmaceuticals-15-01477-f004:**
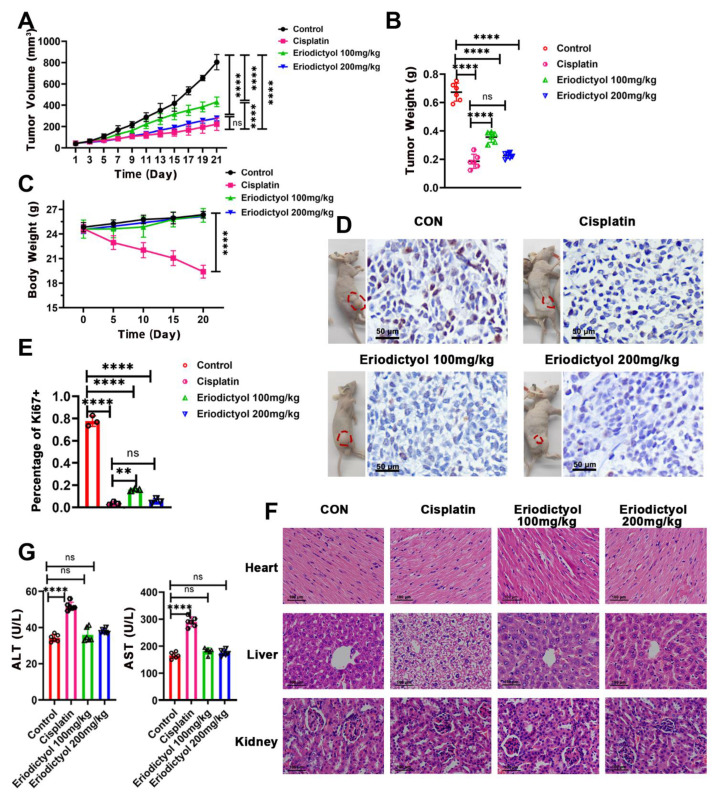
Eriodictyol inhibits tumor growth in vivo. (**A**–**C**) The tumor volumes (**A**), tumor weights (**B**), and body weights (**C**) of the mice from control, cisplatin-treated, and eriodictyol-treated groups were measured, and the statistical results are shown (*n* = 6). (**D**) The representative images of mice bearing tumors and the Ki-67 staining results of the corresponding tumors are shown. Original magnification, 400×; scale bars, 50 µm. (**E**) The quantitative analysis of Ki-67-positive percentages is demonstrated as a histogram. (**F**) H&E staining of major organs from each group. Original magnification, 400×; scale bars, 100 µm. (**G**) Serum levels of ALT and AST were detected and are statistically illustrated as a histogram (*n* = 6). Data are presented as mean ± SD. ns: no significance, ** *p* < 0.01, **** *p* < 0.0001.

**Figure 5 pharmaceuticals-15-01477-f005:**
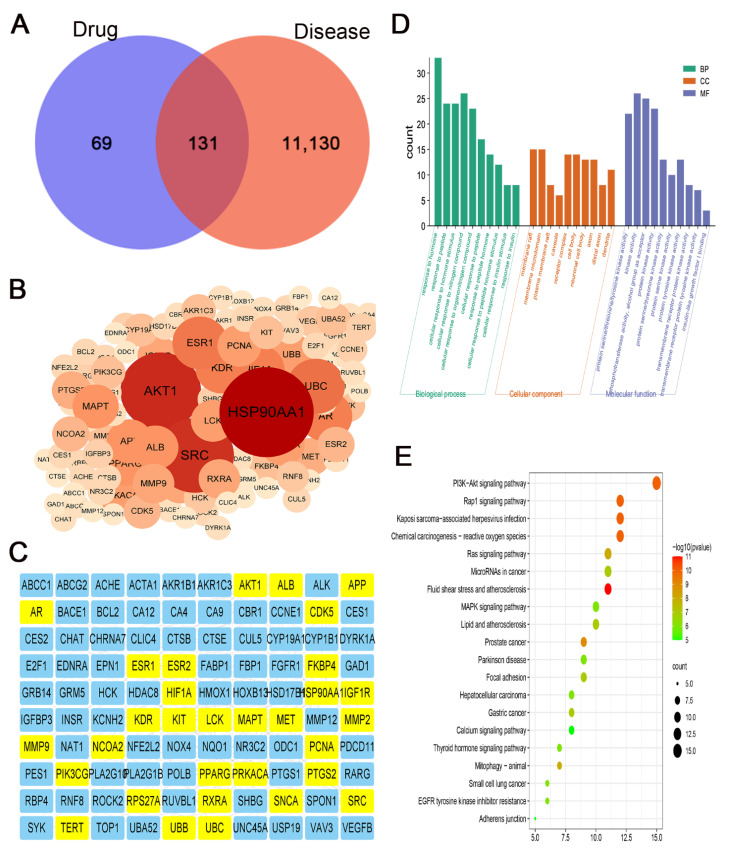
Network pharmacology analysis of the candidate targets of eriodictyol against GC. (**A**) Venn diagram revealing the overlapping potential targets between eriodictyol treatment and GC. (**B**) The result of PPI analysis for overlapping potential targets. (**C**) 100 core targets of eriodictyol acting on GC. (**D**) The result of GO enrichment analysis for the potential pathways of eriodictyol in GC. (**E**) The result of KEGG enrichment analysis for the potential pathways of eriodictyol in GC.

**Figure 6 pharmaceuticals-15-01477-f006:**
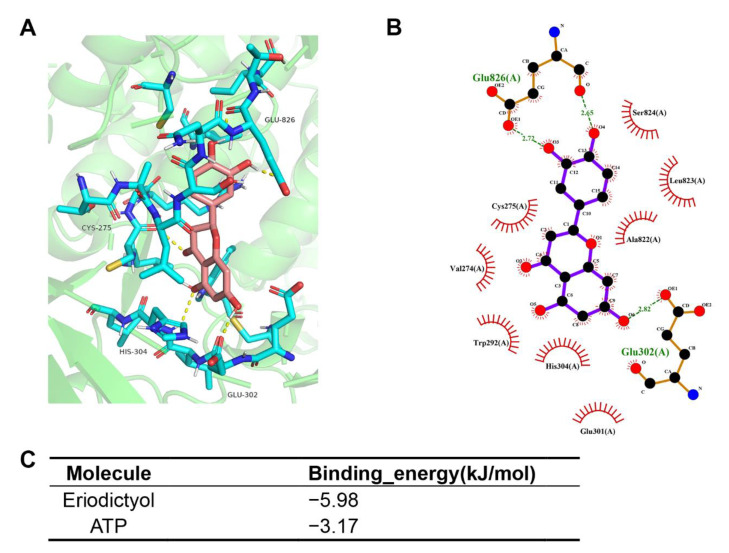
Molecular docking models of eriodictyol and PIK3CG. (**A,B**) Schematic diagram of docking results of 3D (**A**) and 2D (**B**) interactions between eriodictyol and PIK3CG. (**C**) Affinity analysis of eriodictyol and ATP bonding to PIK3CG.

**Figure 7 pharmaceuticals-15-01477-f007:**
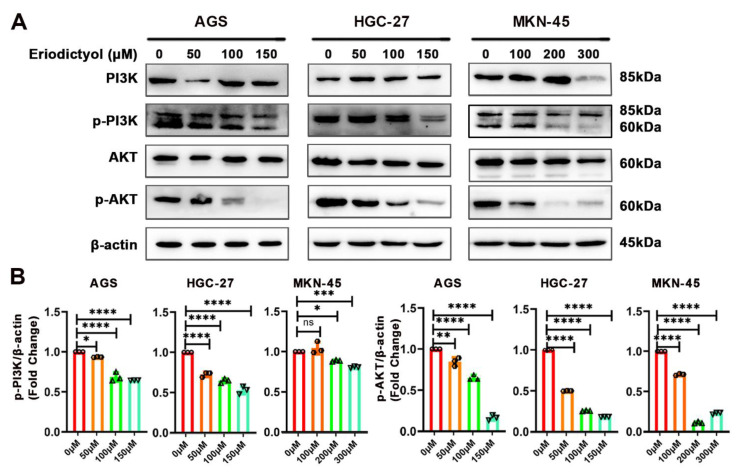
Eriodictyol inhibits GC through the PI3K/AKT pathway. (**A**) The protein levels of PI3K, phosphorylated PI3K, AKT, and phosphorylated AKT were measured with Western blotting. (**B**) The relative protein levels of phosphorylated PI3K and phosphorylated AKT were quantified and are demonstrated as a histogram. Three independent experiments were performed, and data are presented as mean ± SD. ns: no significance, * *p* < 0.05, ** *p* < 0.01, *** *p* < 0.001, **** *p* < 0.0001.

**Figure 8 pharmaceuticals-15-01477-f008:**
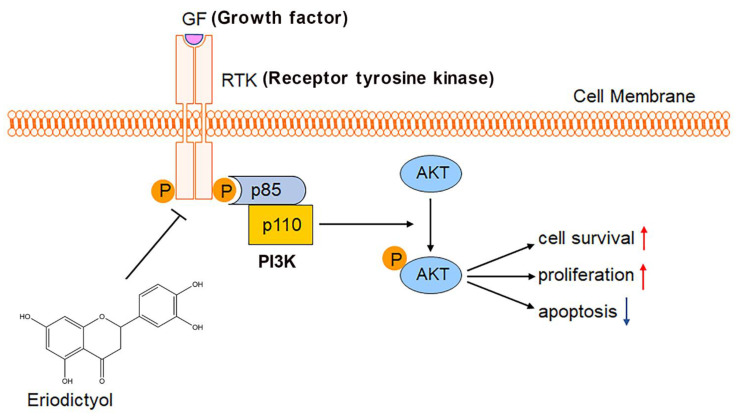
Schematic diagram of the study. Eriodictyol inhibits cell survival and proliferation and induces apoptosis in GC through the PI3K/AKT pathway.

**Table 1 pharmaceuticals-15-01477-t001:** IC50 of eriodictyol of GC cells and gastric epithelial GES-1 cells.

Cell Line	IC50 (µM)		
	24 h	48 h	72 h
AGS	210.5	155.9	110.6
HGC-27	217.0	115.2	41.2
MKN-45	441.5	319.2	216.9
GES-1	955.7	486.1	283.1

## Data Availability

Data are contained within the article.

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
