# Peer review of "Eriodictyol Suppresses Gastric Cancer Cells via Inhibition of PI3K/AKT Pathway"

_pharmaceuticals, 2022, doi:10.3390/ph15121477_

Round 1

Reviewer 1 Report

The manuscript entitled “Eriodictyol suppresses gastric cancer cells via inhibition of PI3K/AKT pathway investigated the anti-cancer effects of eriodictyol towards GC both in vitro and in vivo.

The manuscript is very interesting and well-written. However, minor revisions should be made in order to be published in Pharmaceuticals journal, and the manuscript should be completed and/or modified taking into account the suggestions below:

1.      The authors are advised to rephrase the sentences from lines 37-39, 58-60, 145-146  

2.      The authors are advised to change the following sentence: „Nevertheless, the anti-cancer activity and potential targets of eriodictyol have not been well studied” (lines 62-63), since several papers investigated theese effects (references 29, 31, 33)

3.      The authors are advised to better explain  how they choose the doses of eriodictyol (100-200mg/kg) for 4.7. Xenograft Tumor Model

4.      The authors are advised to add details about the standards used (eriodictyol, quercetin, taxifolin and kaempferol)

Author Response

Please see the attachment。

Reviewer 2 Report

The manuscript "Eriodictyol suppresses gastric cancer cells via inhibition of  PI3K/AKT pathway" reports new findings worthy of publication in Processes. Before publishing the data in Processes, the reviewer thinks several points should be corrected.

Concerns & Suggestions:

1. If a non-tumoral cell line is evaluated to determine the selectivity, authors should also test compounds in five concentrations, to determine the IC50 values at 24 h, 48 h and 72 h.

2.  In the manuscript the authors didn’t mention how to get Eriodictyol and how about the purity? This raises a question mark about it’s purity and hence relevance and reliability of the biological results.
